# Painting-Emotion Matching Technology Learning System through Repetition

**Taemin Lee [1] and Sanghyun Seo [2,\*]**

[1]  Department of Computer Science and Engineering, Chung-Ang University, Seoul 06974, Korea
[2]  School of Computer Art, College of Art and Technology, Chung-Ang University,
    Anseong-si 17546, Kyunggi-do, Korea
\*  Correspondence: sanghyun@cau.ac.kr; Tel.: +82-2-824-3018

**Abstract:** People's interest in paintings has increased as artists have easier access to an audience. However, at times, laypersons may not understand the significance of a painting. With the development of computer science, it has become possible to analyze paintings using machines, but some limitations remain. In this paper, we present a learning tool to help analyze the sensitivity of a given painting. To this end, the proposed system provides users with the ability to predict the emotions expressed by a painting through repeated learning of a matched painting. Using this learning tool, users can improve their ability to understand paintings.

**Keywords:** education system for matching with emotion and painting; emotion extraction from paintings; painting classification based on their emotion

---

## 1. Introduction

As access to paintings by famous artists has become easier, we have more opportunities to admire paintings. Paintings are drawn with certain emotions according to the artist's intention. While some paintings can be easily appreciated, some other paintings are difficult to understand, as shown in Figure 1. People who specialize in painting rarely find it hard to appreciate, but non-experts often feel nothing when they observe these paintings. As the amount of information available for paintings increases, non-experts are more likely to encounter "difficult" paintings. Thanks to advances in computer science, non-experts can now use machines to admire paintings based on current human emotions [1] or search for paintings by emotional adjectives [2]. However, there are limitations to carrying around such a system all the time to admire the painting. If non-experts can learn the emotions in paintings in a simple way, they have the advantage of being able to admire them without the help of other methods. For doctors who provide mental care for patients or for curators who explain and show paintings, this system would help them to easily collect and classify paintings that they can show or recommend. In addition, non-realistic rendering technologies using programs are developing. Rendering systems can also be improved if learning increases the ability to find emotion from paintings [3–6]. Therefore, this study aims to improve the ability of users to understand paintings by providing them with learning tools that help them to analyze the emotions of a given painting.

This paper proposes a system that teaches non-experts to understand the emotions in paintings, by using repeated learning with paintings in which the emotions are labeled. The main purpose of the system is to recommend paintings by using technology-enhanced learning (TEL). The system works as follows: it shows the user one painting and a few other randomly selected paintings. The user chooses a painting that is the most emotionally close to the first given painting, and provides a score according to the selection. The paintings are sourced from a database in which the emotions are labeled in advance. The database contains pre-collected information about emotions in paintings and was

created using a prediction model based on a convolutional neural network (CNN). Each time a user selects a painting, a list of other recommended paintings showing the closest match to his/her choice is provided. The system facilities repeated learning as users can choose the closest matching painting from the recommended options, and shows which emotional groups are finally selected. Thus, users would be able to admire the paintings they choose, learn more about the emotions associated with them, and have the ability to categorize similar emotions.

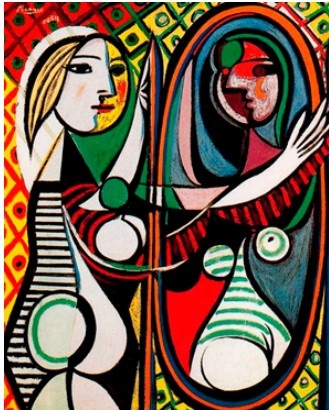

**Figure 1.** Example of a painting with a hard-to-understand emotional message ("Girl before a Mirror," Pablo Ruiz Picasso, 1932).

The main contributions of this study are as follows. First, our system teaches users to understand the emotions in paintings by using repeated emotional classification learning. Secondly, by using a CNN, we constructed a high-performance emotional prediction model that can automatically detect (or predict) emotions in a given painting. Unlike a real photo image that only shows visual information, a painting is drawn by an artist to reflect his or her own emotions. Therefore, a painting-emotion prediction model with a high accuracy is different from previous studies that used photos. Third, we tested the system and verified the accuracy of the recommended lists via a user study. We confirmed that the users could perceive the correct emotions.

The remainder of this paper is structured as follows. Section 2 describes relevant studies on e-learning using multimedia content, such as images and music. It also describes the research on analyzing and extracting emotions from images. Section 3 describes the system that teaches users how to categorize emotions in paintings. In this way, a system that can acquire the ability to predict emotions is described in paintings. Emotional values of the paintings shown are provided by a CNN-based model that predicts emotions. Section 4 presents and validates our results. Finally, we provide a summary of our method, results, and future work.

## 2. Related Work

Various studies on e-learning using content have been carried out. Learning these contents through TEL is very efficient [7]. Content can be largely divided according to one's senses, with auditory elements such as music [8] and visual elements such as pictures and videos. Visual tools have been used to create a system to guide students not well-versed with musical instruments to handle them [9]. The system was verified via scoring. A gesture interface has also been used to support music education [10], wherein a small wireless sensor interface helps users understand music comfortably by analyzing gestures and playing music that matches the gestures. Others [11,12] have proposed a system for creating digital pictures using music. Users choose colors with their feet while playing music, and provide digital pictures that reflect the sounds and colors. This can help others to learn music visually. Thus, many studies have shown that music-based education approaches use the concept of play while listening to music so that users become familiar with the music. Similarly, this



study attempts to provide users easy access to the relationship between paintings and emotions so that they can predict the emotions of paintings.

However, little research has been conducted on TEL in relation to paintings. Lee et al. provided tools to help users easily learn line drawings by analyzing what they are currently drawing when they want to draw an object [13]. This tool allows users to become familiar with drawing and improve their ability to draw. The aim of our study is not to improve users' creative abilities, but to enhance their knowledge of paintings so that they become more adept at understanding and admiring them.

Various studies have extracted emotions from image data. For example, emotions in photographic images have been classified [14,15]. Both studies produced models based on a database called the International Affective Picture System (IAPS), which contained photos and their corresponding emotions, and they focused on values of arousal–valence (AV) [16]. The photographs in the IAPS were grouped according to eight emotional adjectives using a number of training sets. However, the studies were limited with regard to finding the physical distance of proximity in the emotional space (in terms of AV coordinates) as their AV values were classified, rather than predicted, into eight categories. Apart from photographs, studies have been conducted to extract emotions from videos [17], synchronized videos, and music. Each frame of the video used was then analyzed, and an AV value was predicted separately for each frame. Other works have referred to non-paintings as domains [14,15,17], and their methods cannot be applied directly to paintings.

A few studies have extracted emotions from paintings [1,2]. For instance, Kang et al. [2] predicted the emotional adjective of painting by using a three-color combination on the color image scale. After finding the three main colors based on ratio and similarity, the corresponding emotional adjective was mapped to become the representative adjective of the paintings. Furthermore, Lee et al. [1] found the AV value of this adjective, calculated the emotional value of the painting, and mapped it to music. Both the aforementioned studies thus extracted emotions from paintings. However, the results were not highly reliable because they predicted emotions using only combinations of colors. In this study, we overcome this limitation by creating a CNN-based model to predict emotions from the collected paintings as AV values.

## 3. System

In order to develop the ability to predict emotions form paintings, we want users to learn by repeatedly recommending paintings. Our system can be largely divided into three steps; the User-Guide, Learning System, and Building Database. The User-Guide part describes the rules of paintings that the user should be familiar with before using the learning system. This describes features that come from analyzing direct paintings, to provide information that can help users select paintings that are close to each other when choosing a painting in the learning system. In the Learning system phase, the user actually selects the closest painting he thinks is from the painting options shown. Give four different paintings for one painting and choose the closest one emotionally. After selecting one, show and select four other options around the first painting and the selected painting. Through this type of iterative learning, the user is able to learn by showing the score on the user's choice and by showing the last thing about the closest matched set. Building Database produces a model that can predict the emotional value of paintings used in the learning system. Using pre-collected paintings-AV values, an emotion prediction model is developed through CNN. Use this to estimate the AV value for any paintings and to use it as an option in the Learning system. Figure 2 shows our system flow.

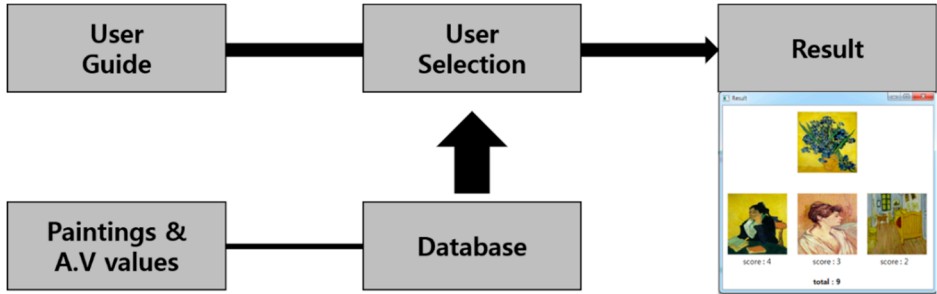

**Figure 2.** System overview.

### 3.1. User Guide

Before starting the learning, users are informed about the basic classification of the paintings. The learning objectives of the users can be divided into three main categories. First, we predefine the emotions expressed in the paintings via categorization. Second, we provide the information collected from existing art books. Third, the information is analyzed using a linear regression approach. The following information is provided to the users.

**Predefined categorization**

1. The size of the painting does not affect the emotion. (In reality, very large paintings may increase the arousal value, but the paintings we examined were not large.)
2. The paintings we collected were captured under similar conditions.

**Information from existing art books [18–21]**

1. The lower the saturation of the painting, the gloomier the image. (This feature corresponds to a small valence value.)
2. The more the use of the color red in the painting, the more excited the user. (This feature corresponds to a big arousal value.)
3. The more varied the direction of the lines, the more confused the user. (This feature corresponds to a big arousal value and a small valence value.)
4. The more the number of vertical lines, the calmer the feeling imparted by the painting. (This feature corresponds to a small arousal value.)

**Information from linear regression analysis**

1. The more diagonal the alignment of the image, the higher its arousal value.
2. The higher the brightness, the higher the arousal value.
3. The higher the symmetry between the left- and right-hand sides of the image, the higher the arousal value.
4. The higher the saturation, the higher the valence value.
5. The higher the saturation, the higher the arousal value.
6. Larger the value that structurally and more symmetrical valence.

To find the relationship between the characteristics of the paintings and the AV values, we used the coefficients of determination provided by the linear regression. The regression was performed by extracting the features from the collected paintings, their AV values, and the paintings themselves. The characteristics used for the paintings, the corresponding coefficients of determination, and some of the features are shown in Table 1. Using the process described above, the User Guide can provide exceptionally large values and visually distinguishable features to users.

**Table 1.** Features with coefficients of determination for estimating emotion. (bigger than 0.60 is useful for estimating).

| Features | Arousal | Valence |
|---|---|---|
| Physical value | Wavelet_LH (0.65), LV (0.21), LD (0.55), AH (0.72), AVE_saturation (0.73), brightness (0.76) | Wavelet_LH (0.71), LV (0.33), LD (0.65), AH (0.32), AVE_saturation (0.63), brightness (0.82) |
| Symmetry | H_Symmetry_Color/Entropy (0.70, 0.13), V_Symmetry_Color/Entropy (0.62, 0.42) | H_Symmetry_Color/Entropy (0.50, 0.70), V_Symmetry_Color/Entropy (0.65, 0.30) |

### 3.2. Learning System

To teach users how to classify the emotions in paintings, repeated lessons are conducted where users view and select paintings. The system shows four paintings, and the user chooses one of them. These paintings are representative, and each of them belongs to a different emotional group. When one of the paintings is selected, four new paintings are shown around it. These four paintings have the nearest, furthest, and two intermediate distances from the first selected painting in the emotional space (AV coordinate), according to the database that contains them. The user chooses a painting among these that he/she thinks is the closest to the original emotion. When the second painting is selected, three other paintings are recommended based on the first selected painting and intermediate point. Likewise, the three paintings are the closest, the furthest, and the middle ones, based on the midpoint. After that, two paintings are recommended and selected based on the focus of the three selected paintings. The final scoring will be done by giving 4 to 1 points for the first choice, 3 to 1 for the second, and 2 to 1 for the last selection. The scope of the score is determined by the number of options shown. This score help to arouse learner's interest. The system shows the paintings that should have been selected for the first representative emotional group and the user's score when the selection is complete. It also describes the common characteristics of paintings in the same emotional group, and concludes the learning. Figure 3 shows the order of learning.

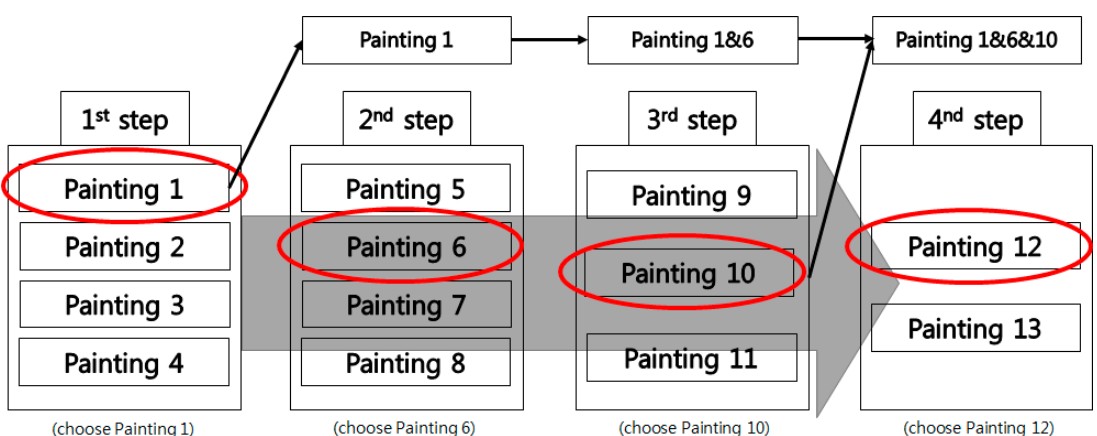

**Figure 3.** Overview of the learning system.

Figure 4 shows the results obtained at the end of all the selection steps seen in Figure 3. Figure 4a is an example of a result when a painting closest in emotion to the first painting is always selected, whereas Figure 4b illustrates a result when the most appropriate matching painting is not selected in the second step.

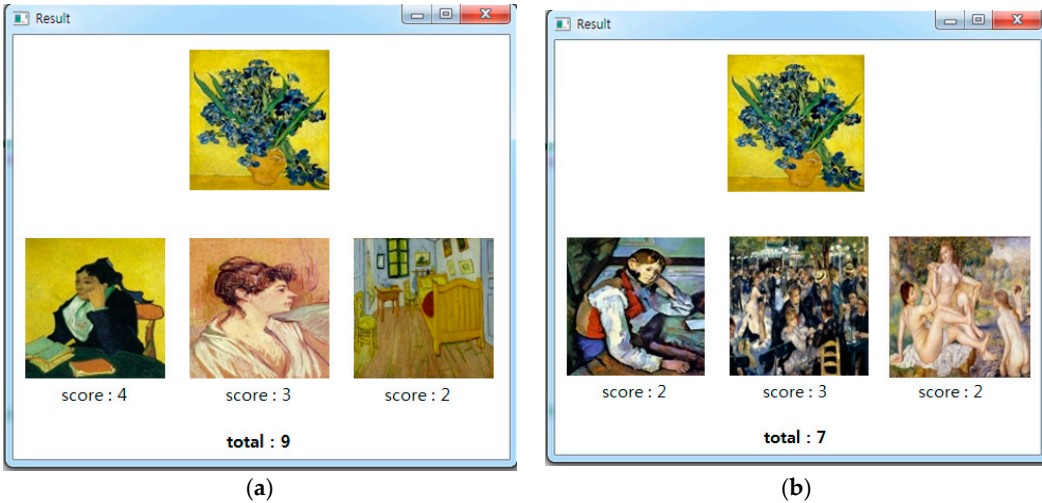

**Figure 4.** Result scene of selection: (**a**) description of selecting all the closest paintings; (**b**) description of selecting another painting in the 2nd step.3.3. Building the Database.

### 3.2.1. Data Collection

To apply CNN, we required three paintings with known values of arousal and valence. The ground-truth value was defined as the correct value by averaging the AV values from ordinary users. To this end, a user questionnaire, as shown in Figure 5, was used. To obtain the average of non-experts' appreciation of paintings, 65 images were shown to 50 non-experts, and the value of AV was assessed in five levels using the SD method. The questions were asked in English to enable the survey to be expanded, and a preliminary explanation was included for cases where the intensity of the emotional axis was not understood, and verbal explanations were also included for a complete understanding. We first showed a dictionary meaning for arousal and valence. The meaning of arousal is as follows: it is the state of being excited. The meaning of valence is as follows: it is the state of being positive. In addition, it was explained that having a large arousal value is activate and small value is deactivate. Also, an example is given to adjectives that normally exist at both ends of the AV model. The adjectives representing the large arousal were described as "excited" and small value is "sleepy". The large valence is pleasant and the small valence is unpleasant. The representative adjective is "satisfied or happy" and the opposite is "disappointed". The survey was conducted for users in their 20s and 30s. Only 57 images out of 65 were included in the database, because variation in survey responses was too large for the remaining images.

### 3.2.2. Data Collection

Using AV values from 1 to 5 collected in Section 3.2.1, we produced ground-truth values. Table 2 shows examples of ground-truth values that we collected and produced. The average of values collected on a 1–5 scale was calculated and rescaled to a 1–9 scale to express the emotion of the painting. An AV value for the painting was defined, as shown in Table 2, to produce a prediction model for emotions using 57 data samples.

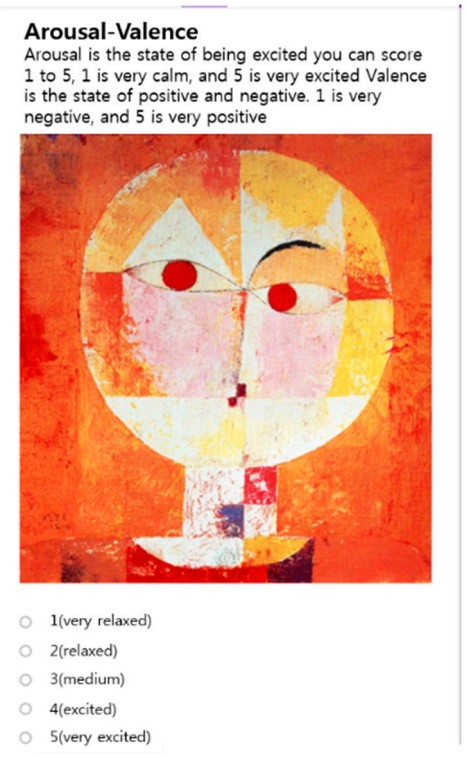

**Figure 5.** User study paper; Let user choose one of the 5 option with their feelings when they see the paintings, options are range from very relaxed to very excited.

**Table 2.** List of paintings with painter, and the ground-truth value of arousal–valence.

| Index | Name of Painting, Painter | Arousal | Valence |
|:---:|:---:|:---:|:---:|
| 1 | Senecio, Paul Klee | 3.9919 | 5.6635 |
| 2 | L'Arlesienne, Vincent van Gogh | 3.7357 | 5.7221 |
| 3 | Girl in front of Mirror, Picasso | 5.0615 | 4.5905 |
| 4 | The Large Bathers, Renoir | 5.3680 | 5.7916 |
| 5 | Portrait of Marcelle, Lautrec | 3.7442 | 5.3913 |
| 6 | April, Denis | 3.8310 | 6.1975 |
| 7 | The Kiss, Klimt | 5.2809 | 5.6885 |
| 8 | Vairumati, Paul Gauguin | 4.7923 | 5.8430 |
| 9 | The Promenade, Claude Monet | 4.1315 | 5.7807 |
| 10 | La Creation de l'homme, Marc Chagall | 4.6053 | 5.8301 |

### 3.2.3. Results of the Model for Predicting Emotions in Paintings

To produce a learning model, CNN training is repeated 1000 times using Keras on InterCore i7, 16 GB RAM and Windows 7 64-bit operating system. It took approximately 24 h to learn the 50 images in size 768 × 768. We use 4 layers to build the CNN. The number of nodes in each layers is (256, 256, 256, 1). We use relu function for activation. Table 3 shows the results: loss rate and accuracy of our model. We used MSE (mean squared error) for calculating and ADAM (adapted moment estimation) as optimizer Score is used to excite learners. We trained 50 data (#1–#50) and test 20 data (#38–#57) for calculating loss value. The loss rate for arousal is 15.3, the loss rate for valence is 11.9, and the average loss rate is 13.6. The accuracy of our predictive model is 86.4%. Figure 6 shows examples of predictions made by our model. The square denotes the ground-truth value, and the circle denotes the predicted value. As the painting shows, the predicted values are close to ground truth.

**Table 3.** Loss rate and accuracy of prediction model.

| Size | Arousal (Loss) | Valence (Loss) | Average (Loss) | Accuracy |
|---|---|---|---|---|
| 768 × 768 (CNN) | 15.3% | 11.9% | 13.6% | 86.4% |

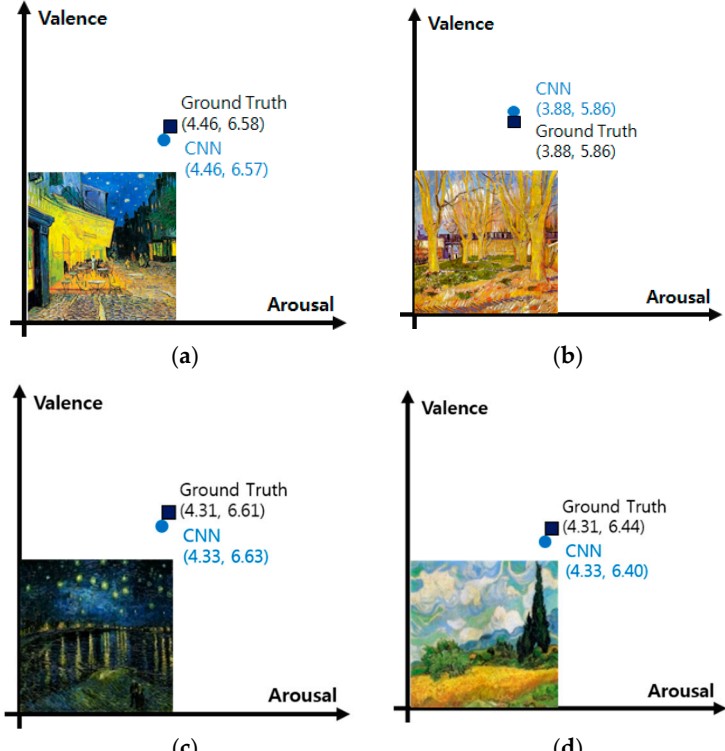

**Figure 6.** Results of convolutional neural network (CNN)-based estimation of emotion in paintings: (**a**) Description of "café terrace", Vincent van Gogh, 1888; (**b**) Description of "Avenue of Plane Trees near Arles Station", Vincent van Gogh, 1888; (**c**) Description of "starry night over the Rhone", Vincent van Gogh 1888; (**d**) Description of "wheat field with cypresses", Vincent van Gogh, 1889.

## 4. Result

### 4.1. Results for the Learning System

Using the proposed system, users repeated learning steps of viewing recommended paintings and scoring them. Five learning sessions were conducted for five users. Figure 7 shows the results of selective and repeated learning of five paintings for one user. The scores for three choices are 1 to 9. For this user, the score was 7.2 on average. This is a wrong score once per full study of the nearest painting choice. Because it is an emotional-based taxonomy, it can be said that the score of 7.2 is well chosen.

Table 4 shows the scores for five attempts with 10 testers. They were all non-professional and could be tested without prejudice. They are all graduate students in their late 20s. For practical learning, the first reference painting should be chosen randomly, but for an accurate comparative analysis, it was defined and tested in advance. The first reference paintings are listed as the first paintings in Figure 7. Table 4 shows results of 5 testers that are part of our test. As shown in Table 4, users 1–5 assigned the scores between 6.0 and 7.4. The average is 7.0 out of 9.0. This means that learning the theory that helps users to categorize paintings before starting to use the learning system is helpful.

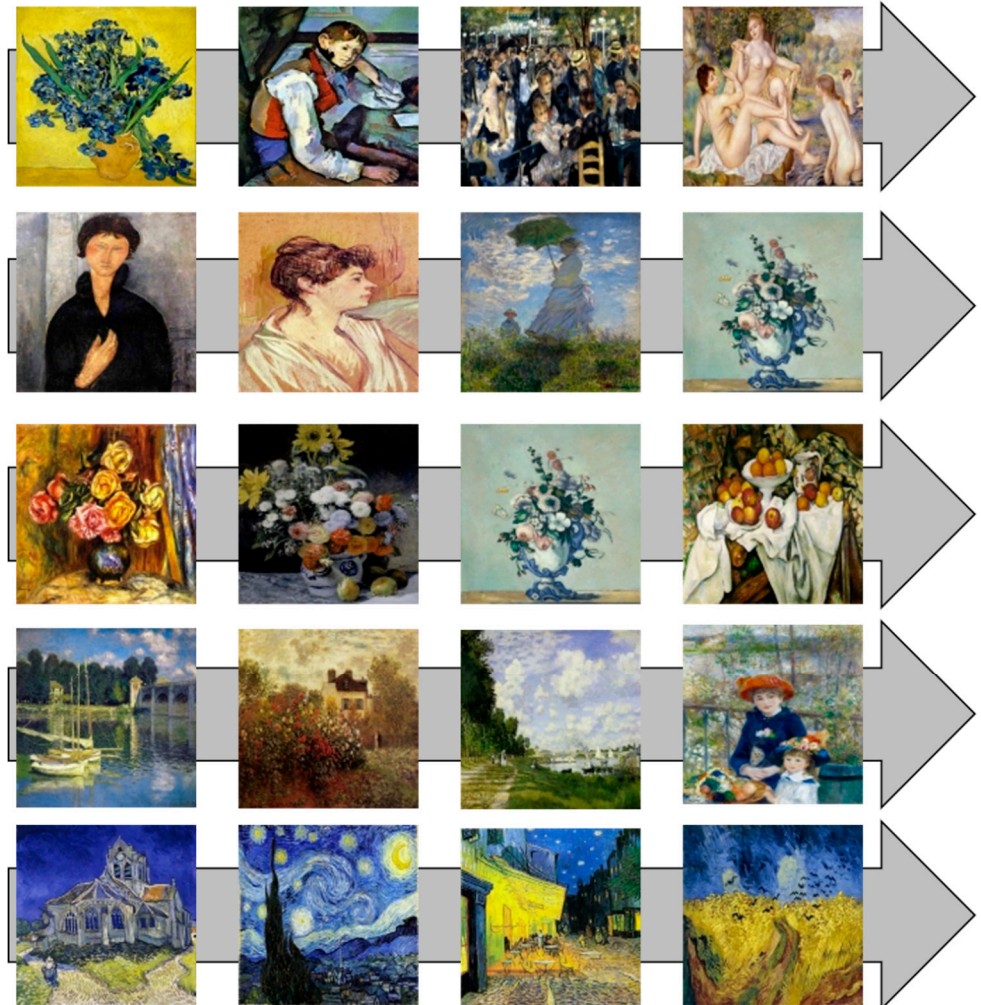

**Figure 7.** Overview of selection process for one user; 1st image is shown painting at the first time, and 2nd image is chosen one which the user think the closest with 1st painting. 3rd image and 4th image is same process with 2nd image. (For the 3rd line, the score is not high. The reason is that other paintings have a closer arousal–valence (AV) value, but people chose the flowers to be similar.

**Table 4.** Data of 5 users (In each attempt, they get the same first painting.).

|          | 1st Attempt | 2nd Attempt | 3rd Attempt | 4th Attempt | 5th Attempt |
|----------|-------------|-------------|-------------|-------------|-------------|
| 1st user | 9           | 6           | 6           | 7           | 8           |
| 2nd user | 9           | 7           | 6           | 6           | 6           |
| 3rd user | 7           | 8           | 5           | 8           | 7           |
| 4th user | 8           | 7           | 9           | 7           | 6           |
| 5th user | 7           | 7           | 9           | 5           | 5           |

## 4.2. Validation of the Learning System

To validate our learning system, we conducted user tests on the process by which they identify similar paintings. Three paintings with similar emotions were shown, and a selection of paintings with the same emotion was made. We conducted two experiments. The first experiment involved viewing and selecting only one painting similar to the first painting. This step checked whether the classification was correct. The second experiment involved viewing several paintings and choosing one of them. Thus, we assessed the accuracy of our learning system by using the scores of emotional distances for each painting. Three points were given to the nearest painting, and one to the farthest painting. The paintings recommended at this time were different from that in the first test, and were

recommended only for those paintings whose emotional distance did not deviate significantly. The system used to show and verify the three closest paintings in emotional terms is depicted in Figure 8.

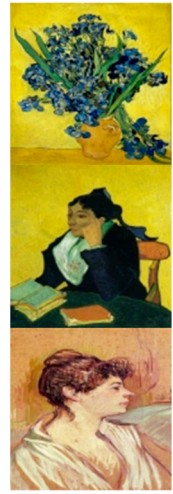 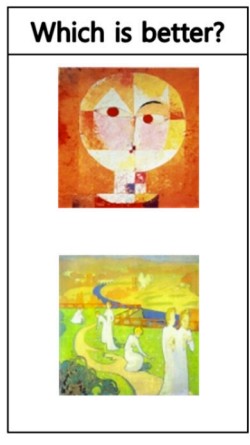 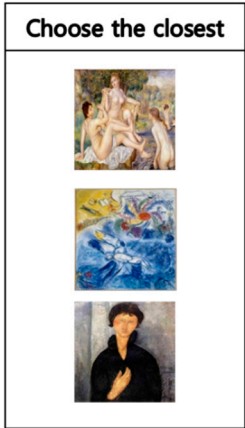

**Figure 8.** Validation system.

As shown in Figure 8, users were evaluated three times. The answer/error result table for selecting one of the two paintings is shown in Table 5. The results for finding the nearest painting among several paintings are shown in Table 6. As shown in Table 5, most of the answers are correct except for user 5. In particular, for users 2, 3, or 4, we can see that the answer has been found. Through this, we can see that users have the ability to distinguish paintings that are emotionally distant from a given painting by using our system.

**Table 5.** Right or wrong answer in finding matched painting.

|  | 1st Test | 2nd Test | 3rd Test |
|---|---|---|---|
| 1st user | correct | incorrect | correct |
| 2nd user | correct | correct | correct |
| 3rd user | correct | correct | correct |
| 4th user | correct | correct | correct |
| 5th user | incorrect | incorrect | correct |
| 6th user | correct | correct | correct |
| 7th user | incorrect | correct | incorrect |
| 8th user | incorrect | correct | correct |
| 9th user | correct | correct | correct |
| 10th user | correct | correct | correct |

**Table 6.** Score of finding closest image (repeated tester).

|  | 1st Test | 2nd Test | 3rd Test | Average |
|---|---|---|---|---|
| 1st user | 3 | 3 | 2 | 2.67 |
| 2nd user | 1 | 3 | 3 | 2.33 |
| 3rd user | 3 | 3 | 3 | 3 |
| 4th user | 3 | 3 | 3 | 3 |
| 5th user | 2 | 2 | 1 | 1.67 |
| 6th user | 3 | 3 | 3 | 3 |
| 7th user | 2 | 2 | 2 | 2 |
| 8th user | 3 | 3 | 1 | 2.33 |
| 9th user | 1 | 2 | 3 | 2 |
| 10th user | 3 | 3 | 3 | 3 |

As shown in Table 6, users scored 2.67, 2.33, 3.00, and 1.67, respectively, in three attempts. The overall average score was 2.5, and it is approximately 84% when recalculated as accuracy. As shown in Table 6, the 5th user does not yet have a sufficient understanding of painting: if we exclude this user, the average score is 2.59 and accuracy is 86.3%. That is, if the learning process goes well using our learning system, users can say that they can classify paintings with approximately 86% accuracy.

To verify the benefits of repeated learning, we conducted tests on three users who did not repeat it. The results of the test are shown in Table 7. One out of three people had good scores, but the other two had poor scores. The user with good scores is also relatively small compared to repeated learners. As we can see from this result, it is more effective to learn paintings.

**Table 7.** Score of finding closest image (non-repeated tester).

|          | 1st Test | 2nd Test | 3rd Test | Average |
|----------|----------|----------|----------|---------|
| 1st user | 2        | 3        | 2        | 2.33    |
| 2nd user | 1        | 3        | 1        | 1.67    |
| 3rd user | 2        | 2        | 2        | 2       |

## 5. Conclusions

By recommending and selecting a painting to the users, this study encourages the user to develop the ability to estimate emotions of paintings. For this, the users are given guidelines for classifying paintings. Apart from common guidelines, we also include the rules obtained by regression. Once the user learns this information, the system shows several paintings (choosing them by using their AV coordinates). The user chooses the painting that they think is closest to the reference painting. By repeating this task several times, the user is evaluated for the accuracy of their chosen paintings. Finally, our learning methods were validated by verifying that the users learned well.

This research makes three major contributions. First, it allows the user to learn to understand emotions in paintings by using repeated learning for emotional classification. As validated in chapter 4, this learning method is helpful to users. Second, by using CNN, we produced a high-performance emotional prediction model that can automatically detect emotions from a given painting. Using machine learning resulted in increasing accuracy to 86.4%.

There are some limitations in our research. First, our training dataset for the predictive model of paintings is limited. With more ground-truth data, CNN can create a more accurate model. To extend the dataset, several people should provide an AV value for each painting. It is necessary to investigate A.V. values of many paintings, and there are practical limitations in conducting a user study. Therefore, it is necessary to establish a system that can easily collect information from social networks. Second, it is necessary to increase the amount of educational information about paintings. There are currently 10 ways to learn about the classification of paintings, but there are more ways to categorize them. Finding more of this information and explaining to the user how to learn will make it easier for users to make choices in the learning system stage. Third, another emotional classification method is needed. The current classification of paintings is called arousal–valence. Because AV coordinates are often used in emotional research, this taxonomy is useful. However, this part may be less intuitive for non-experts. Using more intuitive emotional groups will make it easier for users to understand the material. If this learning is perfectly defined, it is expected that application development can be used to help with emotional therapy.

**Author Contributions:** T.L. and S.S. participated in all phases and contributed equally to this work. T.L. wrote the paper and S.S. revised the whole paper. T.L. performed experimental data collection and S.S. advised in the process of paper-writing.

**Funding:** This research was funded by the Korea government (MSIT) grant number [No. NRF-2019R1F1A10 58715 through the National Research Foundation of Korea (NRF)].

**Acknowledgments:** This work was supported by the National Research Foundation of Korea (NRF) grant funded by the Korea government (MSIT) (NRF-2019R1F1A1058715).

**Conflicts of Interest:** The authors declare no conflict of interest.

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
