# Peer review of "Painting-Emotion Matching Technology Learning System through Repetition"

_sustainability, doi:10.3390/su11164507_

Round 1
Reviewer 1 Report
The paper discusses an interesting idea, which is matching emotions with paintings and using machine learning models to predict emotions of paintings. It also attempts to build a system that can teach non-expert users how to valuate paintings based on the Arousal/valence labels. The idea is very interesting but very challenging as well as there are many points that need more grounding in the literature of art and painting interpretations which I believe the paper simplified or omitted to a great extent. The writing and structure of the paper needs to be improved. Here is some detailed comments:
- The paper writing and structure is not clear and needs to be improved. In section 3, the first paragraph is still about related work, why this appears under "system" ? The user guide and learning system sections seem to be misplaced and described before the dataset or the regressions system is introduced, which makes the paper hard to read. Many English and writing typos, for example, the paper ends with an incomplete sentence at the end of the conclusion section. Also Tables and Figures labelling needs to be revised, Figure 3 is meant to be figure 4. Table 4 is mentioned in text, but it is not there.
- Many theories in the paper needs more grounding and references from the literature of arts and understanding paintings. I am not an expert in this area but, for example, terms like "sensitivity of a given paining" is not clear and ambiguous. The user guide section mentions rules regarding evaluating the paintings, what is the basis of these rules? What are the theories these rules are extracted from?
- Evaluating paintings is very subjective. The labelling of the database seem to be taken from non-expert labellers. It is not clear how many annotations are collected per one painting. No inter-rater agreement values were reported to show the reliability of these labels. As mentioned in the paper, the Arousal/Valence concepts are not intuitive for non-expert users, so this raises questions about the reliablility of the used labels. More details about labelling need to be added and inter-rater agreement analysis need to be included as well, something like Kappa or Krippendorff's alpha
- The CNN used for predicting is explained very birefly, more details about the architecture, the hyper-parameter tuning and dataset divisions (training, development, and testing) are needed. The network classifies two values, for arousal and valence, what is the the architecture used ? is this a multitask network ? Aslo, the dataset is very small to train a deep network, it would be useful to try simple machine learning models, such as SVM to see how it will perform before using a CNN
- The user studies reported at the end need more details in describing the procedure. Also it seems that only 5 users were recruited for the user studies, which is a very small number.
I believe the paper has an interesting idea but needs more work to be in publishable state.
Author Response
1) The paper writing and structure is not clear and needs to be improved. In section 3, the first paragraph is still about related work, why this appears under "system" ?
Answer: We revised our writing. The modified parts are as follows:
|
In order to develop the ability to predict emotions form paintings, we want users to learn by repeatedly recommending paintings. Our system can be largely divided into three steps; the User-Guide, Learning System, Building Database. User-Guide part describes the rules of paintings that the user should be familiar with before using the learning system. Describes features that come from analyzing direct paintings, to provide information that can help users select close paintings when choosing a painting in the learning system. In the Learning system phase, the user actually selects the closest painting he thinks is from the painting options shown. Give four different paintings for one painting and choose the closest one emotionally. After selecting one, show and select four other options around the first painting and the selected painting. Through this type of iterative learning, the user is able to learn by showing the score on the user’s choice and by showing the last thing about the closest matched set. Building Database produces a model that can predict the emotional value of paintings used in the learning system. Using pre-collected paintings-A.V. values, an emotion prediction model is developed through CNN. Use this to estimate the A.V. value for any paintings and to use it as an option in the Learning system.
|
The user guide and learning system sections seem to be misplaced and described before the dataset or the regressions system is introduced, which makes the paper hard to read. Many English and writing typos, for example, the paper ends with an incomplete sentence at the end of the conclusion section. Also Tables and Figures labelling needs to be revised, Figure 3 is meant to be figure 4. Table 4 is mentioned in text, but it is not there.
Answer: I revised all errors.
|
Figure 3 (a) -> Figure 4 (a) Figure 3 (b) -> Figure 4 (b) Table 5 -> Table. 4 Table 6 -> Table. 5 Table 7 -> Table. 5
|
2) Many theories in the paper needs more grounding and references from the literature of arts and understanding paintings. I am not an expert in this area but, for example, terms like "sensitivity of a given paining" is not clear and ambiguous.
Answer: "Emotion" and "Sensitivity" mentioned in this paper have the same meaning. Since it is not a particular term, both of them were used together as a word to describe a person's feelings. I'm sorry we've confused you. Therefore, all terms were unified into one, "Emotion.".
The user guide section mentions rules regarding evaluating the paintings, what is the basis of these rules? What are the theories these rules are extracted from?
Answer: There are three types of information provided at the User Guide level. 1 and 2 are assumptions in our analysis of paintings. 3-6 provides information previously used in the theory of colors and lines. References to this section are added(reference 18~21). 7-12 provides the most easily visible information from Table 1's correlation coefficient.
3) Evaluating paintings is very subjective. The labelling of the database seem to be taken from non-expert labelers. It is not clear how many annotations are collected per one painting. No inter-rater agreement values were reported to show the reliability of these labels.
Answer: I agree that predicting and evaluating the sensitivity of a conversation can be subjective. While one's emotions cannot all be the same, most of them have sensations that are commonly predicted. We decided to make the assessment on this point. When you import emotional data, you can collect it from experts. But once knowledge is in, we think it can happen that you don't accept the painting as it is. Therefore, the data collected by non-experts were assumed to be an answer set and the study was conducted.
As mentioned in the paper, the Arousal/Valence concepts are not intuitive for non-expert users, so this raises questions about the reliability of the used labels.
Answer: As mentioned in the paper, there is no expected to be a problem that users do not understand, as we have described Arousal and Valence in detail. It also improved data reliability by excluding extreme data when collecting data.
More details about labelling need to be added and inter-rater agreement analysis need to be included as well, something like Kappa or Krippendorff's alpha.
Answer: Thank you for providing us with a way to ensure the reliability of the label. We believed that if investigated by many people, we could gain credibility by doing so. This was not expected to be a big problem, as studies that were not used in other studies also exist. But if we can improve our reliability by using Kappa or Krippendorff's alpha, we will need this. We'd like to add it to our research, but we haven't had time to include it all at the moment. We will make sure to use your proposed method to improve the reliability of labeling later on.
4) The CNN used for predicting is explained very briefly, more details about the architecture, the hyper-parameter tuning and dataset divisions (training, development, and testing) are needed. The network classifies two values, for arousal and valence, what is the the architecture used? is this a multitask network ?
Answer: We use for 4 layers to build CNN with (256,256,256,1) nodes. We use “relu” function for activation. In actual programming, the following code is used.
|
model = Sequential() model.add(Dense(256, activation='relu',input_dim = width*height*3)) model.add(Dense(256, activation='relu')) model.add(Dense(256)) model.add(Dense(1)) |
We used MSE(mean squared error) for calculating and ADAM(Adapted Moment Estimation) as optimizer Score is used to excite learners. We trained 50 data(#1-#50) and test 20 data(#38-#57) for calculating loss value.
|
To produce a learning model, CNN training is repeated 1,000 times using Keras on InterCore i7, 16GB RAM and Windows 7 64-bit operating system. It took approximately 24 hours to learn the 50 images in size 768*768. We use for 4 layers to build CNN. The number of nodes in each layers is (256, 256, 256, 1). We use relu function for activation. Table 3 shows the results: loss rate and accuracy of our model. We used MSE(mean squared error) for calculating and ADAM(Adapted Moment Estimation) as optimizer Score is used to excite learners. We trained 50 data(#1-#50) and test 20 data(#38-#57) for calculating loss value. The loss rate for arousal is 15.3, the loss rate for valence is 11.9, and the average loss rate is 13.6. The accuracy of our predictive model is 86.4%.
|
Also, the dataset is very small to train a deep network, it would be useful to try simple machine learning models, such as SVM to see how it will perform before using a CNN
Answer: Less learning data can be unreliable in performing CNN. Because it is difficult to build a database that matches pictures and emotions, there is less learning data. As the number of data increases, CNN's results will increase confidence. Although CNN's limitations exist today, we think the study's theme is the possibility of producing emotional paintings through CNN and the ability to learn from them. Emotional therapy through the application will allow us to expect the database to expand with the feedback from it and solve the limitations.
We used linear regression analysis for estimating emotion from paintings. For this, we need to define the features and extracted them from paintings. The features are subjective, and have low accuracy than CNN. If there is no big difference when comparing accuracy, CNN, which is easy to expand without having to select features, was considered to be advantageous. Below is our result of comparing CNN and Linear regression
5) The user studies reported at the end need more details in describing the procedure. Also it seems that only 5 users were recruited for the user studies, which is a very small number.
Answer: We later learned five more people and reflected them in the results. The results of the five added persons are as follows.
|
1st test2nd test3rd testAverage1st user3322.672nd user1332.333rd user33334th user33335th user2211.676th user33337th user22228th user3312.339th user123210th user3333
|
Also we added a description of the score and table 7 to explain the benefits of repeat learning. Table 7 shows test results of non-repeating learners. This proved that it is effective to repeat learning. The added parts are as follows:
|
To verify the benefits of repeated learning, we conducted tests on three users who did not repeat it. The results of the test are shown in Table 7. One out of three people had good scores, but the other two had poor scores. The user with good scores is also relatively small compared to repeated learners. As we can see this results, it is more effective to learn paintings. Table 7. Score of finding closest image(non-repeated tester) 1st test2nd test3rd testAverage1st user2322.332nd user1311.673rd user2222
|

Reviewer 2 Report
This paper proposes a method to predict the emotion that can be felt in painting based on CNN, and to use this model to train people to recognize the emotions of painting. The intent of this work is understandable to some extent, and it seems that some experiments have been carried out to support the conclusion. However, it seems necessary to revise the paper in consideration of the following several points.
1) It seems necessary to clarify more clearly why human training is needed in this work. There is a brief mention of the field of application related to art therapy in the introduction. It is necessary to appeal such practical reasons in abstract. The difficulty of simply recognizing the emotions embedded in paintings by non-professionals is hardly a motive for this work.
2) Many places emphasize "repeated learning". How does this differ from "non-repeated learning"? It is necessary to compare the results of two cases through experiments.
3) There seems to be a lack of explanation as to how the score obtained by the user during repeated training is calculated. This score plays a key role in showing the results of the experiment, so it would be better to explain it more intuitively in the previous section.
4) This work builds a model that predicts the emotion inherent in painting based on CNN, and it seems necessary to clarify some points. This work has implemented CNN with 57 image data, which is typically too small for the number of other CNN models that deal with images. Can you say that you have overcome these limitations enough? How can you say it is possible if you have overcome it? In addition, there is no mention of the structure of the CNN-based overall model used, that is, the number of layers and the number of nodes, and there is also no explanation as to why such a structure is used. How was the accuracy of the model calculated (Line 220)? How is the training data and testing data differentiated in the calculation of the accuracy?
The following are minor errors.
Line 186 Figure 3 (a) -> Figure 4 (a)
Line 187 Figure 3 (b) -> Figure 4 (b)
Line 242: There is no Table 4
Line 248: We have only Table 5.
Author Response
1) It seems necessary to clarify more clearly why human training is needed in this work. There is a brief mention of the field of application related to art therapy in the introduction. It is necessary to appeal such practical reasons in abstract. The difficulty of simply recognizing the emotions embedded in paintings by non-professionals is hardly a motive for this work.
Answer: When non-specialists observe a painting, they say it is simply beautiful, grand and so on. Although intuitive paintings exist, they often fail to define how they feel when they see them. As you mentioned, it is expected to be of great help as it is applied to Art Teraphy. Thank you for your suggestion for informing us of the development of our research. However, there was a time limit to add experimental studies to help with emotional therapy. Therefore, we referred to the application development using it in conclusion section(last sentence). The added part is as follow:
|
If this learning is perfectly defined, it is expected that application development can be used to help with emotional therapy.
|
2) Many places emphasize "repeated learning". How does this differ from "non-repeated learning"? It is necessary to compare the results of two cases through experiments.
Answer: In some cases, a user can acquire it right away from a single learning session. But in general, I think repeat learning can better educate users. To show this, we added Table 7, once comparing scores through learning. Three people were tested. Two had bad score and one had good score. The added table is as follows.
|
To verify the benefits of repeated learning, we conducted tests on three users who did not repeat it. The results of the test are shown in Table 7. One out of three people had good scores, but the other two had poor scores. The user with good scores is also relatively small compared to repeated learners. As we can see this results, it is more effective to learn paintings. Table 7. Score of finding closest image(non-repeated tester) 1st test2nd test3rd testAverage1st user2322.332nd user1311.673rd user2222
|
3) There seems to be a lack of explanation as to how the score obtained by the user during repeated training is calculated. This score plays a key role in showing the results of the experiment, so it would be better to explain it more intuitively in the previous section.
Answer: Score is used to excite learners. It is intended to make users learn as if they were playing a game by scoring the choice of pictures. The range of Score is 1 to 4, 1 to 3 in the second selection, and 1 to 2 in the third selection. The scope of the score is determined by the number of options shown. So, with a total score of nine points, it helps users acquire effectively by showing them the scores of the pictures they find. Rather than play an important role in experimenting with results, we think it is important to arouse learners' interest. So this part is added to Section 3.2.
4) This work builds a model that predicts the emotion inherent in painting based on CNN, and it seems necessary to clarify some points. This work has implemented CNN with 57 image data, which is typically too small for the number of other CNN models that deal with images. Can you say that you have overcome these limitations enough? How can you say it is possible if you have overcome it?
Answer: Less learning data can be unreliable in performing CNN. Because it is difficult to build a database that matches pictures and emotions, there is less learning data. As the number of data increases, CNN's results will increase confidence. Although CNN's limitations exist today, we think the study's theme is the possibility of producing emotional paintings through CNN and the ability to learn from them. As you mentioned above, emotional therapy through the application will allow us to expect the database to expand with the feedback from it and solve the limitations.
In addition, there is no mention of the structure of the CNN-based overall model used, that is, the number of layers and the number of nodes, and there is also no explanation as to why such a structure is used.
Answer: We use for 4 layers to build CNN with (256,256,256,1) nodes. We use “relu” function for activation. In actual programming, the following code is used.
|
model = Sequential() model.add(Dense(256, activation='relu',input_dim = width*height*3)) model.add(Dense(256, activation='relu')) model.add(Dense(256)) model.add(Dense(1)) |
How was the accuracy of the model calculated (Line 220)? How is the training data and testing data differentiated in the calculation of the accuracy?
Answer: We used MSE(mean squared error) for calculating and ADAM(Adapted Moment Estimation) as optimizer Score is used to excite learners. We trained 50 data(#1-#50) and test 20 data(#38-#57) for calculating loss value. This parts is added in our paper.
|
To produce a learning model, CNN training is repeated 1,000 times using Keras on InterCore i7, 16GB RAM and Windows 7 64-bit operating system. It took approximately 24 hours to learn the 50 images in size 768*768. We use for 4 layers to build CNN. The number of nodes in each layers is (256, 256, 256, 1). We use relu function for activation. Table 3 shows the results: loss rate and accuracy of our model. We used MSE(mean squared error) for calculating and ADAM(Adapted Moment Estimation) as optimizer Score is used to excite learners. We trained 50 data(#1-#50) and test 20 data(#38-#57) for calculating loss value. The loss rate for arousal is 15.3, the loss rate for valence is 11.9, and the average loss rate is 13.6. The accuracy of our predictive model is 86.4%.
|
5) The following are minor errors.
Line 186 Figure 3 (a) -> Figure 4 (a)
Line 187 Figure 3 (b) -> Figure 4 (b)
Line 242: There is no Table 4
Line 248: We have only Table 5.
Answer: I revised all errors.

Reviewer 3 Report
It is a good way to educate users to analyze pictures. The analysis of the figures may vary, but it is likely to be applicable in many studies as long as the emotional values used in this paper can be found properly. I think it would be nice to make more technological advances so that they can be applied more in real life.
People's interest in paintings has increased as artists have easy access to them. But there are paintings that are hard to understand. With the development of computer science, it is also possible to analyze paintings through machines, but it is not easy to analyze them with a machine whenever you observe them. This paper provides people with learning tools to analyze the sensitivity of a given painting. To do this, t propose a system that learns the matched picture over and over again to give the user the ability to predict emotions in the picture. This learning tool can help users improve their ability to understand pictures.
There are few suggestions for improving this paper. First of all, recent researches that are related with TEL. The main topic of this journal is TEL. So more relation with TEL is necessary to be mentioned The research that are studied within five years. If there is no research related with your research, other researches with TEL are needed to be added. Secondly, the result of estimating emotion from painting is added. I think showing various results will effectively assert your paper. Lastly, it is necessary to refer to profile of people for user study.
Author Response
1) Recent researches that are related with TEL. The main topic of this journal is TEL. So more relation with TEL is necessary to be mentioned The research that are studied within five years. If there is no research related with your research, other researches with TEL are needed to be added.
Answer: I think music and painting are easy contents to learn through TEL. There has been a lot of TEL learning about music[Leman and Nijs2016]. On the other hand, there is a study of drawing related to the TEL, but there is no study of the TEL in learning the painting. So we add two related works with TEL. ; [16,17] is added
Reference : Leman, M., and L. Nijs. 2016. “Cognition and Technology for Instrumental Music Learning.” In The Routledge Companion to Music, Technology & Education, edited by A. King, , 23–36. London: Routledge
Erik Duval, Mike Sharples, and Rosamund Sutherland. Research themes in technology enhanced learning. In Technology Enhanced Learning, pages 1-10. Springer, 2017
2) The result of estimating emotion from painting is added. I think showing various results will effectively assert your paper.
Answer: We added two more results of estimating emotion from paintings. We revised Figure 6. The added results are as follows:
| (a) (b)(c) (d)
Figure 6. Results of CNN-based estimation of emotion in paintings: (a) Description of “café terrace”, Vincent van Gogh, 1888; (b) Description of “Avenue of Plane Trees near Arles Station”, Vincent van Gogh, 1888; (c) Description of “starry night over the Rhone”, Vincent van Gogh 1888 ;(d) Description of “wheat field with cypresses”, Vincent van Gogh, 1889
|
3) It is necessary to refer to profile of people for user study.
Answer: We referred about profile of people for user study. The number of users is 50 and they are all non-experts in paintings and 20~30s. However, we didn’t referred about the profile of testers. They were all non-professional and could be tested without prejudice. They are all graduate students in their late 20s. We added this part to the paper. Added parts are as follows:
|
Table 4 shows the scores for five attempts with 10 testers. They were all non-professional and could be tested without prejudice. They are all graduate students in their late 20s. For practical learning, the first reference painting should be chosen randomly, but for an accurate comparative analysis, it was defined and tested in advance. The first reference paintings are listed as the first paintings in Figure 7. Table 4 shows results of 5 testers that are part of our test. |

Round 2
Reviewer 1 Report
Thanks to the authors for adressing most of my comments and the paper has clearly improved. Few more points that was not addressed:
-Inter-rater agreement is just a statistical test to show that the labels are accurate. It is simple to calculate as many tools are available if the sample labels are still available.
-How did the authors explained arousal and valence to the participants, more details about the experimental procedure will be useful
Author Response
1) Inter-rater agreement is just a statistical test to show that the labels are accurate. It is simple to calculate as many tools are available if the sample labels are still available.
Answer: First of all we thank you for your advice. In general, we thought we could improve our credibility by getting rid of extreme data by being investigated by people. I couldn't use the various methods you said right away, but I'm feeling the need. Although we have looked at several things during the short revised time, we expect more research to be needed on this part. In future studies, we will add this to improve reliability. Once again, thank you for your advice on the reliability of the data.
2) How did the authors explained arousal and valence to the participants, more details about the experimental procedure will be useful
Answer: We helped the user understand through the meaning of the dictionary, the meaning of the two coordinates, and the representative adjectives. This section has been added as follows to revise our paper.
|
To apply CNN, we required three paintings with known values of arousal and valence. The ground-truth value was defined as the correct value by averaging the A.V. values from ordinary users. To this end, a user questionnaire, as shown in Figure 5, was used. To obtain the average of non-experts' appreciation of paintings, 65 images were shown to 50 non-experts, and the value of A.V. was assessed in five levels using the SD method. The questions were asked in English to enable the survey to be expanded, and a preliminary explanation was included for cases where the intensity of the emotional axis was not understood, and verbal explanations were also included for a complete understanding. We first showed a dictionary meaning for arousal and valence. The meaning of arousal is as follows: It is the state of being excited. The meaning of valence is as follows: It is the state of positive. In addition, it was explained that having a large arousal value is activate and small value is deactivate. Also, an example is given to adjectives that normally exist at both ends of the A.V. model. The adjectives representing the large arousal were described as “excited” and small value is “sleepy”. The large valence is pleasant and the small valence is unpleasant. The representative adjective is “satisfied or happy” and the opposite is “disappointed”. |
